# Systematic Review and Network Meta-Analysis of Anaplastic Lymphoma Kinase (ALK) Inhibitors for Treatment-Naïve ALK-Positive Lung Cancer

**DOI:** 10.3390/cancers13081966

**Published:** 2021-04-19

**Authors:** Cheng-Hao Chuang, Hsiao-Ling Chen, Hsiu-Mei Chang, Yu-Chen Tsai, Kuan-Li Wu, I-Hua Chen, Kung-Chao Chen, Jui-Ying Lee, Yong-Chieh Chang, Chin-Ling Chen, Yu-Kang Tu, Jen-Yu Hung, Chih-Jen Yang, Inn-Wen Chong

**Affiliations:** 1Division of Pulmonary and Critical Care Medicine, Department of Internal Medicine, Kaohsiung Medical University Hospital, Kaohsiung Medical University, Kaohsiung 807, Taiwan; 1040239@kmuh.org.tw (C.-H.C.); 1010362@kmuh.org.tw (Y.-C.T.); 1070476@kmuh.org.tw (K.-L.W.); jyhung@kmu.edu.tw (J.-Y.H.); 2Department of Pharmacy, Kaohsiung Municipal Ta-Tung Hospital, Kaohsiung Medical University Hospital, Kaohsiung Medical University, Kaohsiung 807, Taiwan; 1058065@kmuh.org.tw (H.-L.C.); 880504@kmhk.org.tw (H.-M.C.); 980770@kmuh.org.tw (Y.-C.C.); 3Department of Internal Medicine, Kaohsiung Medical University Hospital, Kaohsiung Medical University, Kaohsiung 807, Taiwan; 1060473@kmuh.org.tw (I-H.C.); 1060475@kmuh.org.tw (K.-C.C.); 4Division of Chest Surgery, Department of Surgery, Kaohsiung Medical University Hospital, Kaohsiung Medical University, Kaohsiung 807, Taiwan; rockwell@cc.kmu.edu.tw; 5Cancer Center, Kaohsiung Medical University Hospital, Kaohsiung Medical University, Kaohsiung 807, Taiwan; 960645@kmuh.org.tw; 6Institute of Epidemiology and Preventive Medicine, National Taiwan University, Taipei 100, Taiwan; yukangtu@ntu.edu.tw; 7Department of Medical Research, National Taiwan University Hospital, Taipei 100, Taiwan; 8Faculty of Medicine, College of Medicine, Kaohsiung Medical University, Kaohsiung 807, Taiwan; 9Department of General Medicine, Kaohsiung Medical University Hospital, Kaohsiung Medical University, Kaohsiung 807, Taiwan; 10Respiratory Therapy, College of Medicine, Kaohsiung Medical University, Kaohsiung 807, Taiwan

**Keywords:** network meta-analysis, crizotinib, alectinib, ceritinib, brigatinib, ensartinib, lorlatinib, ALK inhibitor

## Abstract

**Simple Summary:**

The prognosis of non-small cell lung cancer (NSCLC) is poor as more than half of the patients are diagnosed at an advanced stage. The outcomes have greatly improved in the recent two decades due to the introduction of target therapy. Anaplastic lymphoma kinase (ALK) rearrangement is the second most common driver mutation found in lung cancer, and several ALK inhibitors have shown excellent efficacy. However, head-to-head randomized controlled trials of these agents are lacking. We conducted this systematic review and network meta-analysis to indirectly compare the currently available ALK inhibitors. In summary, lorlatinib had the highest probability of the best progression-free survival in first-line treatment of ALK-positive NSCLC, followed by low-dose (300 mg twice daily) alectinib, high-dose (600 mg twice daily) alectinib, brigatinib, ensartinib, and ceritinib. Lorlatinib was associated with the best objective response rate and the highest probability of grade 3–5 adverse effects. Low-dose alectinib had the best safety profile.

**Abstract:**

Several anaplastic lymphoma kinase inhibitors (ALKIs) have demonstrated excellent efficacy on overall survival (OS), progression-free survival (PFS), objective response rate (ORR), and also better adverse effect (AE) profiles compared to cytotoxic chemotherapy in advanced stage anaplastic lymphoma kinase (ALK) rearrangement-positive non-small cell lung cancer (NSCLC) in phase III randomized clinical trials (RCTs). We conducted this systematic review and network meta-analysis to provide a ranking of ALKIs for treatment-naïve ALK-positive patients in terms of PFS, ORR, and AEs. In addition, a sub-group analysis of treatment benefits in patients with baseline brain metastasis was also conducted. Contrast-based analysis was performed for multiple treatment comparisons with the restricted maximum likelihood approach. Treatment rank was estimated using the surface under the cumulative ranking curve (SUCRA), as well as the probability of being the best (Prbest) reference. All next-generation ALKIs were superior to crizotinib in PFS but lorlatinib and brigatinib had increased AEs. The probability of lorlatinib being ranked first among all treatment arms was highest (SUCRA = 93.3%, Prbest = 71.8%), although there were no significant differences in pairwise comparisons with high- (600 mg twice daily) and low- (300 mg twice daily) dose alectinib. In subgroup analysis of patients with baseline brain metastasis, low-dose alectinib had the best PFS (SUCRA = 87.3%, Prbest = 74.9%). Lorlatinib was associated with the best ranking for ORR (SUCRA = 90.3%, Prbest = 71.3%), although there were no significant differences in pairwise comparisons with the other ALKIs. In addition, low-dose alectinib had the best safety performance (SUCRA = 99.4%, Prbest = 97.9%). Lorlatinib and low-dose alectinib had the best PFS and ORR in the overall population and baseline brain metastasis subgroup, respectively. Low-dose alectinib had the lowest AE risk among the available ALKIs. Further head-to-head large-scale phase III RCTs are needed to verify our conclusions.

## 1. Introduction

Non-small cell lung cancer (NSCLC) accounts for 85% of all lung cancers and is further classified into major subtypes, including adenocarcinoma, squamous carcinoma, and large cell carcinoma [1,2]. Tremendous advances in NSCLC treatment have been made in the past two decades, contributing to improvements in knowledge with regards to the mechanism of driver mutations and cancer biology, with the corresponding development of therapies including small molecular tyrosine kinase inhibitors, antiangiogenesis agents, and immune checkpoint inhibitors [3]. However, lung cancer remains the leading cause of cancer death worldwide, with approximately 1.8 million diagnoses and 1.6 million deaths every year despite the considerable progress in diagnostic and treatment modalities [4,5,6]. A high prevalence of advanced-stage disease at the initial diagnosis may partly explain the poor prognosis, with 55% of patients diagnosed with metastatic disease and a five-year survival rate of only 6.1% [7,8]. How best to care for patients with advanced NSCLC is a crucial clinical challenge, and ongoing efforts are essential.

Some targetable driver mutations have been identified, with the most common types being epidermal growth factor receptor (EGFR) mutations and anaplastic lymphoma kinase (ALK) rearrangements, which are in general mutually exclusive. The discovery of these targetable genetic alterations has led to lung cancer treatments ranging from traditional cytotoxic therapy to the era of precision medicine with personalized small molecular inhibitors. Among these genetic driver mutations, as many as 5%–8% of NSCLC patients have ALK rearrangements, and mostly in younger never smokers [9]. The ALK gene encodes a transmembrane receptor tyrosine kinase with unclear function, and several types of aberrant rearrangements with other genes located on different chromosomes have been reported [10]. Among these rearrangements in lung cancer, the echinoderm microtubule-associated protein-like 4 (EML4) gene located on chromosome 2 is the most common and was first reported in 2007 by Soda et al. [10,11]. ALK–EML4 fusion protein results in overactive signaling with the consequent up-regulation of cell growth, proliferation, survival, and ultimately tumor formation [12].

ALK rearrangements are often associated with younger age, never smokers, and tendency of brain metastasis, which confers a poor prognosis. However, excellent survival benefits of ALK inhibitors (ALKIs) for patients with advanced NSCLC have been demonstrated in several large clinical trials [13,14,15,16,17,18]. Crizotinib was the first ALKI to be approved by the FDA for advanced-stage NSCLC in 2015, and progression-free survival (PFS) and objective response rate (ORR) benefits were demonstrated in head-to-head comparisons with platinum-based chemotherapy in the PROFILE 1014 trial (median PFS 10.9 vs. 7.0 months, HR = 0.45, *p* < 0.001; ORR = 74% vs. 45%) [14]. However, the high incidence of secondary mutations in the ALK gene during crizotinib treatment limits its clinical application. In addition, the next-generation ALKI, ceritinib, demonstrated significantly longer PFS and ORR compared to platinum-based chemotherapy in untreated ALK-positive advanced lung cancer patients in the ASCEND-4 trial (median PFS 16.6 vs. 8.1 months, HR = 0.55, *p* < 0.00001; ORR = 72.5% vs. 26.7%) [15]. Due to their superior PFS, ORR, and even overall survival (OS), ALKIs have become the standard first-line treatment for advanced ALK-positive NSCLC.

Patients treated with first-line crizotinib invariably experience progression due to secondary ALK gene mutations and transformed cancer clones adapting to the therapy, thereby conferring resistance. Recently, several next-generation ALKIs, including alectinib, brigatinib, ensartinib, and lorlatinib have been developed and have been shown to have different spectrum against ALK gene mutations [10]. In addition to their ability to overcome specific ALK secondary mutations, the new generation of ALKIs, including ceritinib, alectinib, brigatinib, and ensartinib have all demonstrated considerable response rates and survival benefits in crizotinib-refractory groups in phase II studies [19,20,21,22]. Due to the efficacy of ALKIs, several large-scale phase III randomized controlled trials (RCTs) have been conducted to determine the efficacy and safety of the frontline use of these new ALKIs in treatment-naïve ALK-positive disease, and most have shown better clinical efficacy, central nervous system penetration and safety profile [13,15,16,17,18,23]. In the ALEX trial, alectinib had a significant PFS advantage over crizotinib (median PFS 34.8 vs. 10.9 months, HR 0.43) in untreated ALK-positive patients with advanced NSCLC with similar safety outcomes [13]. In Asian populations, the efficacy of alectinib was also shown to be superior to crizotinib in two large RCTs, including the J-ALEX trial, which used a half dose (300 mg) of alectinib (median PFS 34.1 vs. 10.2 months, HR = 0.37, *p* < 0.0001) and the AlESIA trial (median PFS not reached vs. 10.7 months, HR = 0.37) [17,18,24]. Brigatinib was also demonstrated to provide better survival than crizotinib in ALKI-naïve patients in the ALTA-1L trial (median PFS 24.0 vs. 11.0 months, HR = 0.49, *p* < 0.0001), although a mild increase in ≥grade 3 adverse effects (AEs) was noted (73% vs. 61%) [16]. A newly developed second-generation ALKI, ensartinib, also exhibited better PFS than crizotinib in recently released interim data [23]. Lorlatinib, a potent and brain penetrant third-generation ALKI was originally only used to treat patients with refractory secondary ALK mutations, including the G1202R solvent front mutation. However, its efficacy and advantage over crizotinib as first-line systemic therapy for patients with advanced ALK-positive NSCLC was demonstrated in the CROWN study (median PFS not reached vs. 9.3, HR = 0.28, *p* < 0.001) [25]. Therefore, these new-generation ALKIs have become the first-line therapy for ALK-positive treatment-naïve patients, and they have all shown superior clinical efficacy to crizotinib. Nevertheless, head-to-head comparisons among these new ALKIs are lacking. The purpose of this study was to perform a network meta-analysis of currently available phase III RCTs to evaluate which ALKIs are most beneficial for ALK-positive patients in terms of PFS, response rate, and grade 3–5 adverse events (AEs) as major outcomes.

## 2. Results

### 2.1. Literature Search

A total of 1380 studies were identified from a comprehensive literature search. After removing duplicate studies, we screened 918 studies based on their abstracts. Among them, 889 were excluded due to irrelevance, and the six remaining studies were included for qualitative synthesis and quantitative meta-analysis. The PRIMSA flow diagram is presented in Figure 1.

### 2.2. Study Characteristics and Quality Evaluation

The study characteristics of the six included phase III RCTs are presented in Table 1. All of them were conducted for patients with ALK-positive NSCLC not previously treated with ALKIs. Head-to-head comparisons were performed between crizotinib and new-generation ALKIs. The ALTA-1L trial investigated brigatinib, the CROWN trial investigated lorlatinib, the eXalt3 trial investigated ensartinib, and the remaining three RCTs investigated alectinib with a high (600 mg twice daily in the ALESIA and ALEX trials) or low (300 mg twice daily in the J-ALEX trial) dose. The median age was 49–60 years, the percentage of male patients across the trials ranged from 37% to 55%, and 14% to 42% of the included subjects had brain metastasis at baseline.

The results of the quality assessment are presented in Appendix A. Detailed information about eXalt3 was obtained from the protocol at ClinicalTrials.gov as limited data were provided in the report from the 2020 annual meeting of the World Conference on Lung Cancer (WCLC). There was a high risk of performance bias due to the open-label design in all of the included trials. However, the risk of detection bias was lower because a blinded independent review committee was responsible for assessing disease progression or treatment response. Quality was unclear with regards to sequence generation, allocation concealment, and selective reporting due to the lack of detailed information.

### 2.3. Pooled Results for Generation Differences in Efficacy and Safety

The pooled results of generation difference in efficacy and safety are provided in Appendix B. Compared to the first-generation ALKI (crizotinib), new-generation ALKIs significantly increased the PFS (hazard ratio (HR) = 0.41, 95% confidence interval (CI) = 0.34–0.49) and ORR (response ratio = 1.16, 95% CI = 1.09–1.23) but did not lead to a higher incidence of grade 3–5 AEs (risk ratio = 0.86, 95% CI = 0.63–1.18). However, there was high heterogeneity in safety comparisons (I2 = 85%). The patients who received brigatinib or lorlatinib had a higher risk of grade 3–5 AEs than those who received crizotinib (risk ratio = 1.19, 95% CI = 1.00–1.40 for brigatinib, risk ratio = 1.27, 95% CI = 1.07–1.52 for lorlatinib); however, the patients who received alectinib had a better safety profile (risk ratio = 0.66, 95% CI = 0.46–0.94).

### 2.4. Efficacy and Safety Evaluation from the Network Meta-Analysis

Figure 2 presents the network constructions of eligible comparisons. In order to understand the dose-effect of alectinib, high (600 mg twice daily) and low (300 mg twice daily) doses were regarded as different interventions. For PFS and ORR, six interventions were included in the network constructions: crizotinib, brigatinib, lorlatinib, ensartinib, and high- and low-dose alectinib. Due to a lack of information, ensartinib was not included in the network constructions for safety and subgroup analysis. The effect sizes of pairwise comparisons are summarized in Figure 3, with SUCRA rankings in Appendix C and probability of being the best treatment in Appendix D for all efficacy indicators. Furthermore, pairwise safety comparisons are provided in Figure 4, with SUCRA rankings for safety in Appendix E and probability of being the safest treatment in Appendix F. League tables summarizing pairwise comparisons are shown in Appendix G.

#### 2.4.1. Progression-Free Survival

The new-generation ALKIs had superior effects compared to crizotinib (HR = 0.52, 95% CI = 0.36–0.75 for ensartinib, HR = 0.49, 95% CI = 0.35–0.69 for brigatinib, HR = 0.34, 95% CI = 0.21–0.54 for low-dose alectinib, HR = 0.41, 95% CI = 0.32–0.54 for high-dose alectinib, HR = 0.28, 95% CI = 0.19–0.41 for lorlatinib). In comparisons between the new-generation ALKIs, lorlatinib had a noticeable benefit over ensartinib and brigatinib (HR = 0.54, 95% CI = 0.32–0.92 for ensartinib, HR = 0.57, 95% CI = 0.34–0.95 for brigatinib), but a similar effect was demonstrated between lorlatinib and alectinib (HR = 0.82, 95% CI = 0.45–1.51 for low-dose alectinib, HR = 0.68, 95% CI = 0.42–1.08 for high-dose alectinib). In addition, low-dose alectinib was not inferior to high-dose alectinib in PFS (HR = 1.22, 95% CI = 0.72–2.07). Moreover, lorlatinib had the best PFS with highest SUCRA and Prbest values (SUCRA = 93.3%, Prbest = 71.8%), followed by low-dose alectinib (SUCRA = 76.8%), high-dose alectinib (SUCRA = 57.9%), brigatinib (SUCRA = 38.9%), ensartinib (SUCRA = 33.2%), and crizotinib (SUCRA = 0.00%).

Regardless of the status of baseline brain metastasis, the new-generation ALKIs provided a greater PFS than crizotinib. Although there were no significant differences among the new-generation ALKIs in most pairwise comparisons, a lower risk was observed for lorlatinib compared to brigatinib in the patients who did not have baseline brain metastasis (HR = 0.49, 95% CI = 0.27–0.91). Although lorlatinib had the best overall performance in PFS in the included patients, it was associated with a worse PFS compared to low-dose alectinib in the patients with baseline brain metastasis (HR = 2.51, 95% CI = 0.28–22.37). Accordingly, low-dose alectinib had the highest SUCRA and Prbest values in the patients with baseline brain metastasis (SUCRA = 87.3%, Prbest = 74.9%), followed by lorlatinib (SUCRA = 67.3%), brigatinib (SUCRA = 49.9%), high-dose alectinib (SUCRA = 45.4%), and crizotinib (SUCRA = 0.1%). On the other hand, lorlatinib had the best PFS the patients who did not have baseline brain metastasis (SUCRA = 89.4%, Prbest = 66.6%), followed by low-dose alectinib (SUCRA = 70.1%), high-dose alectinib (SUCRA = 61.9%), brigatinib (SUCRA = 28.2%), and crizotinib (SUCRA = 0.4%).

#### 2.4.2. Overall Response Rate

Lorlatinib and alectinib had better ORRs than crizotinib (response ratio (RR) = 1.31, 95% CI = 1.11–1.55 for lorlatinib, RR = 1.13, 95% CI = 1.03–1.24 for high-dose alectinib, RR = 1.16, 95% CI = 1.02–1.32 for low-dose alectinib), but there were no significant differences between the other ALKIs and crizotinib (RR = 1.18, 95% CI = 0.99–1.40 for brigatinib, RR = 1.11, 95% CI = 0.95–1.31 for ensartinib). Although no significantly superior effects were found, lorlatinib had a relatively better ORR than the other ALKIs (RR = 1.18, 95% CI = 0.94–1.48 for ensartinib, RR = 1.11, 95% CI = 0.88–1.42 for brigatinib, RR = 1.13, 95% CI = 0.92–1.39 for low-dose alectinib, RR = 1.16, 95% CI = 0.96–1.40 for high-dose alectinib). Additionally, lorlatinib was associated with the best ranking for ORR (highest SUCRA and Prbest, SUCRA = 90.3%, Prbest = 71.3%), followed by brigatinib (SUCRA = 61.0%), low-dose alectinib (SUCRA = 56.7%), high-dose alectinib (SUCRA = 46.4%), ensartinib (SUCRA = 42.9%), and crizotinib (SUCRA = 2.8%).

#### 2.4.3. Grade 3–5 Adverse Events

Lorlatinib was associated with a higher risk compared to crizotinib, low-dose alectinib and high-dose alectinib (risk ratio = 1.27, 95% CI = 1.07–1.52 for crizotinib, risk ratio = 2.52, 95% CI = 1.67–3.81 for low-dose alectinib, risk ratio = 1.62, 95% CI = 1.24–2.12 for high-dose alectinib). However, there was no significant difference between lorlatinib and brigatinib (risk ratio = 1.07, 95% CI = 0.84–1.37). Alectinib (both high and low dose) had a better safety profile than the other new-generation ALKIs and crizotinib. Furthermore, the risk of grade 3–5 AEs with alectinib treatment increased with a higher dose (risk ratio = 1.56, 95% CI = 1.02–2.38). A higher SUCRA value indicates a lower risk of AEs. Accordingly, low-dose alectinib had the best safety performance (SUCRA = 99.4%, Prbest = 97.9%), followed by high-dose alectinib (SUCRA = 75.3%), crizotinib (SUCRA = 49.3%), brigatinib (SUCRA = 18.6%), and lorlatinib (SUCRA = 7.3%).

## 3. Discussion

ALKIs have been shown to be superior to traditional cytotoxic chemotherapy in ORR, OS, and PFS for treatment-naïve ALK-positive lung cancer patients in several trials. However, no head-to-head comparison RCT has yet been conducted to illustrate superiority among these next-generation ALKIs. Our network meta-analysis provides updated information and a proposed ranking for PFS, OS, ORR, and grade 3–5 AEs for different ALKIs for treatment-naïve ALK-positive lung cancer patients. We believe this study provides reliable and feasible information to help physicians and ALK-positive lung cancer patients when making treatment decisions.

In order to understand the efficacy and AEs of these new-generation ALKIs, several meta-analyses have been performed to identify the most beneficial ALKIs for ALK-positive lung cancer patients. For example, a recent meta-analysis showed that all ALKIs (crizotinib, ceritinib, alectinib, and brigatinib) were superior to chemotherapy in PFS, and that both alectinib and brigatinib demonstrated significantly longer PFS compared to crizotinib and ceritinib [26]. However, this network meta-analysis enrolled patients who had been treated with chemotherapy and ALKIs, which complicated the heterogeneity of the study population and hindered the application into real-world practice. In addition, two large RCTs which enrolled patients who received ensartinib or lorlatinib compared to a crizotinib arm in treatment-naïve ALK-positive patients have recently been reported [23,25]. Nevertheless, a validated network meta-analysis to rank ALKIs as first-line treatment in patients with ALK-positive advanced NSCLC with regards to PFS, OS, ORR, and safety concerns is urgently needed.

In our study, lorlatinib demonstrates the longest PFS with the highest SUCRA and Prbest values. However, in pairwise comparisons, lorlatinib was superior in PFS compared to ensartinib and brigatinib, but there was no significant difference between lorlatinib, or high- and low-dose alectinib. Moreover, low-dose alectinib was counterintuitively associated with a better SUCRA of PFS than high-dose alectinib in both ALK-positive patients with or without brain metastasis, although there was no significant difference in pairwise comparisons. Furthermore, low-dose alectinib demonstrated a better SUCRA of PFS than lorlatinib in ALK-positive patients with brain metastasis. In a previous network meta-analysis including the ALEX, J-ALEX, and ALTA-1L trials, alectinib was found to be superior to brigatinib in patients without brain metastases but inferior to brigatinib in those with brain metastases in terms of PFS SUCRA. The presence of the dimethylphosphine oxide (DMPO) group in brigatinib was hypothesized to contribute to its high central nervous system (CNS) efficacy [27]. Surprisingly, low-dose alectinib, but not high-dose alectinib, was superior to brigatinib even in the patients with baseline CNS metastasis in our network meta-analysis. The exaggerated superior PFS benefits in both the J-ALEX and ALESIA trials, with hazard ratios of 0.08 and 0.11, respectively, in the patients with baseline brain metastases, may partially explain the result of our analysis. Of note, wide 95% confidence intervals, especially 0.01~0.61 of the hazard ratio in the J-ALEX trial, may have led to the variability in the results. On the other hand, whether the Asian populations in the J-ALEX and ALESIA trials demonstrated better ORR, PFS, OS, and CNS response under ALKI is an interesting issue, and further investigations are warranted to investigate the background immune status, tumor microenvironment, and variation of ALK gene rearrangements between Asian and Western populations. Further large-scale RCTs are needed to illustrate the dose-response relationship of alectinib. Ensartinib was inferior to the other ALKIs in PFS in the overall population, and there were no data about brain metastasis reported in the related conference abstract.

With regards to ORR, lorlatinib still ranked the best, followed by brigatinib, low-dose alectinib, high-dose alectinib, and ensartinib. However, no significant differences were found in pairwise comparisons among the involved ALKIs. According to our results, lorlatinib seems to be the most beneficial first-line ALKI when considering treatment efficacy. Of note, data on median OS are often unavailable in current studies, which have demonstrated extraordinary efficacy in prolonging survival with ALKIs. Further updated outcome reports of these RCTs need to be followed to determine the effect of each ALKI on OS, as this may revise decisions with regards to the choice of first-line ALKIs.

Adverse effects often limit the clinical utility of target therapy. Severe adverse effects (SAEs) were reported in more than 25% of ALKI-treated patients in a previous meta-analysis (crizotinib 38.09%, alectinib 26.24%, ceritinib: 41.44%, and brigatinib 41.68%) [28]. Adverse effects clearly play an important role in ALKI selection. In our study, alectinib demonstrated the most favorable safety outcomes, followed by crizotinib and brigatinib, and significant differences were observed in pairwise comparisons. Lorlatinib, although highly potent, was associated with the most prominent grade 3–5 AEs. In addition, low-dose alectinib had a trend of lower risk of AEs than high-dose alectinib, although no significant difference in pairwise comparison, which further emphasizes the urgent need of large-scale RCTs to compare different races and dosages of alectinib. Of note, different ALKIs demonstrated different characteristics in regard to SAEs. Respiratory SAEs were most prominent with ceritinib (14.17%) and brigatinib (13.48%), and CNS SAEs were also more common with ceritinib (8.84%) and brigatinib (7.40%), partly due to their CNS penetrant ability [28]. Although lorlatinib demonstrated excellent efficacy in our study, it was associated with a high rate of SAEs. In a phase III RCT, 72% of the patients who received lorlatinib experienced grade 3–4 SAEs compared to 56% in those who received crizotinib. However, most of the SAEs in the lorlatinib group were manageable and not life-threatening metabolic side effects (hypercholesterolemia 16%, hypertriglyceridemia 20%, and increased weight 17%), and the treatment discontinuation rate was similar (7% in lorlatinib vs. 9% in crizotinib) due to SAEs. It is important for practitioners to choose the most appropriate ALKI according to the AE profile and individual patient characteristics.

There are several limitations to our meta-analysis. First, heterogeneity may exist among the enrolled RCTs due to the study protocols, patient baseline characteristics, and response evaluation bias. Only Asian populations were enrolled in the J-ALEX and ALESIA studies, which may overemphasize the impact of races difference. Over-simplification of the study results may be a drawback of indirect comparisons. However, our study focus on ALKI treatment naïve patients would further decrease heterogeneity compared to prior published network meta-analysis. Second, OS information was lacking in the enrolled RCTs of ALK-positive patients, which prevents the extrapolation of our treatment ranking to long-term outcomes. Third, although only ALKI treatment-naïve population trials were enrolled in the network meta-analysis, there were still discrepancies in prior baseline chemotherapy and brain radiotherapy. Further subgroup analysis had been planned but deferred due to unavailable detailed information about baseline chemotherapy or radiotherapy patient in individual study. Fourth, no available next-generation sequencing information was provided in the enrolled studies to assess the genetic aberrancy, including the type of ALK rearrangement and subsequent secondary mutations following AKLI treatment. Therefore, our results only serve as a platform for future trials on new-generation ALKIs as first-line therapy for advanced ALK-positive NSCLC patients, not as direct evidence to promote any single agent as a frontline option at present.

## 4. Materials and Methods

This study followed the PRISMA (Preferred Reporting Items for Systematic reviews and Meta-analyses) extension statement for network meta-analysis [29]. PROSPERO registration number was CRD42021228647 https://www.crd.york.ac.uk/PROSPERO/display_record.php?RecordID=228647 (accessed on 3 February 2021)

### 4.1. Search Strategy and Study Selection

We identified eligible studies through a comprehensive search of PubMed, Embase, Cochrane Library, and Clinical-Trials.gov up to 12 December 2020 with no language limitation. In order to obtain the latest information, we searched the abstracts in the main oncology congresses databases, including the American Society of Clinical Oncology (ASCO), the American Association for Cancer Research (AACR), the European Society for Medical Oncology (ESMO), and the World Conference on Lung Cancer (WCLC). We also reviewed the reference lists of the retrieved studies to include more relevant studies. The detailed search strategy is presented in Appendix H, and the search keywords were: non-small cell lung cancer (NSCLC), lung adenocarcinoma, squamous cell carcinoma (SCC), large cell lung carcinoma (LCC), anaplastic lymphoma kinase inhibitors (ALKIs), and tyrosine kinase inhibitors (crizotinib, ceritinib, alectinib, brigatinib ensartinib, or lorlatinib). The inclusion criteria were as follows: (1) completed phase II–III RCTs involving adults with advanced or metastatic NSCLC; (2) RCTs which focused on patients with ALK-positive NSCLC not previously treated with ALKIs; (3) RCTs which provided comparisons of efficacy and safety between different ALKIs.

### 4.2. Data Extraction and Quality Assessment

Two independent reviewers (H.L. Chen and C.H. Chuang) screened the articles and performed data extraction and quality assessment. Any unresolved discrepancies in data extraction or appraisal of the results were resolved by discussion with a third reviewer (CJ Yang). The extracted information included trial details, including trial name, published year, phase, previous treatment, baseline characteristics, treatment arm with the number of subjects, and primary or secondary endpoints. Quality assessment was performed using the risk of bias (ROB) assessment tool as recommended by the Cochrane Handbook for Systematic Reviews of Interventions [30,31]. Bias was assessed as a judgment (high, low, or unclear) for individual elements from six domains to address the bias in selection, performance, detection, attrition, reporting, and others.

### 4.3. Data Synthesis and Statistical Analysis

Treatment efficacy was evaluated according to PFS and ORR. Additionally, subgroup analysis was conducted to compare the treatment effect on PFS in patients with or without brain metastasis. The safety outcomes focused on grade 3–5 AEs based on the Common Terminology Criteria for Adverse Events [32]. For PFS, adjusted HR was regarded as the effect size. For binary indicators, the response ratio was used as the indicator for ORR, and risk ratios were used for AEs.

We first performed a pairwise meta-analysis using Review Manager Version 5.3 [33] to compare the efficacy and safety between first-generation (crizotinib) and new-generation ALKIs. A DerSimonian and Laird random-effects model was used, assuming that a common intervention effect was relaxed and that the effect sizes had a normal distribution [34]. I2 statistics were used to measure the proportion of total variation in study estimates attributed to heterogeneity. We then generated network graphs for different outcomes separately to clarify which treatments were compared directly or indirectly. Network meta-analysis was conducted using the frequentist framework with mvmeta Stata command (version 16, Stata, College Station, TX, USA) [35], which has been described in detail in previous studies. Contrast-based analysis was performed for multiple treatment comparisons using the restricted maximum likelihood approach. In a contrast-based model, a treatment contrast, such as A versus B, was used as an observation unit in the network map [36,37]. To split the unit, different parameters were used to contrast A with B in RCTs containing both A and B and in other RCTs. These parameters were estimated jointly within the same model to evaluate the difference [38]. Fixed-effect models were then used, and the same true effect size was assumed for all trials in our study. We used fixed-effect models since, in most cases, the treatment of interest was evaluated in one trial, and the number of included trials per comparison was too small to estimate between-study heterogeneity. Finally, we calculated the SUCRA to rank all of the included ALKIs. The SUCRA is a numeric presentation of the overall ranking and presents a single number associated with each treatment. SUCRA values range from 0 to 100%, with a larger value indicating a better rank of the intervention effects. Ranking based on SUCRA accounts better for the uncertainty in estimated treatment effects. Treatment with a higher SUCRA value indicated a better rank of the intervention effect and lower risk of grade 3–5 AEs.

## 5. Conclusions

In summary, our network meta-analysis showed that all next-generation AKLIs have longer PFS and ORR compared to crizotinib, while lorlatinib ranked first in the SUCRA ranking analysis followed by low-dose alectinib, high-dose alectinib, brigatinib, and ensartinib. In pairwise comparisons, lorlatinib was superior in PFS compared to ceritinib and brigatinib, but there was no significant difference compared to low-dose and high-dose alectinib. In a subgroup analysis of baseline brain metastasis population, low-dose alectinib had the highest ranking of PFS, followed by lorlatinib, brigatinib, and high dose alectinib. A counterintuitive finding was that low-dose alectinib demonstrated a better survival benefit than high-dose alectinib in both the overall population and baseline brain metastasis patients. Of note, all data on low-dose alectinib came from Asian populations, and occult race and genetic factors need to be considered in the interpretation of the results. Low-dose alectinib had the best ranking of safety in terms of grade 3–5 adverse effects, followed by high-dose alectinib, crizotinib, brigatinib, and lorlatinib. Given the limited number of studies in the meta-analysis, additional large-scale phase III CT studies are needed to verify these conclusions.

## Figures and Tables

**Figure 1 cancers-13-01966-f001:**
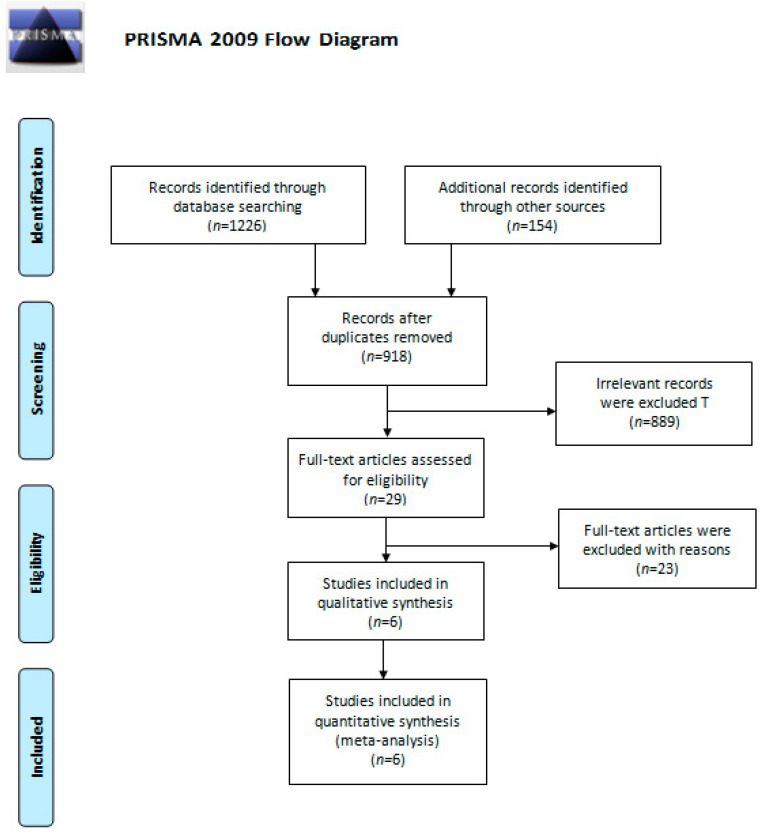
The PRIMSA flow diagram.

**Figure 2 cancers-13-01966-f002:**
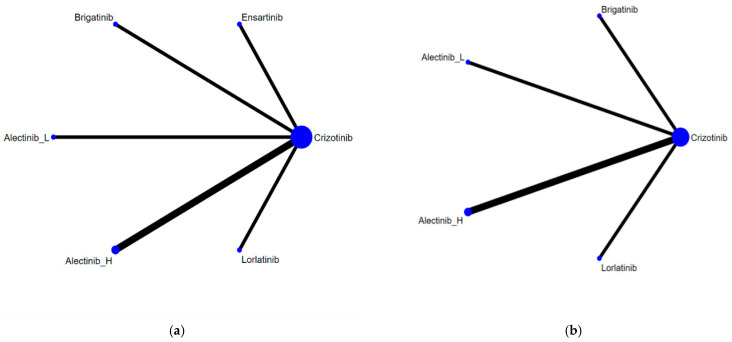
Network constructions for comparisons in PFS, ORR, and grade 3–5 AEs: (**a**) Network constructions for PFS and ORR; (**b**) Network constructions for PFS subgroup analysis and grade 3–5 AEs.

**Figure 3 cancers-13-01966-f003:**
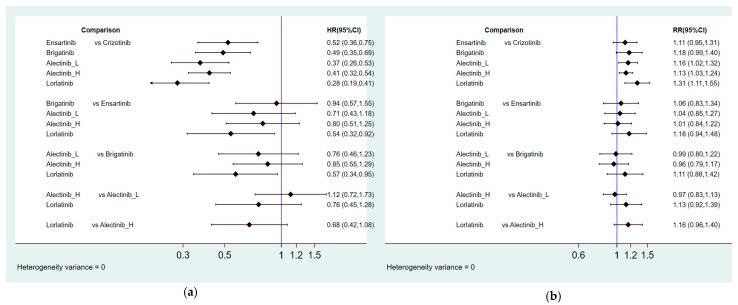
Summary of effect sizes for efficacy comparison: (**a**) Pairwise comparisons for PFS; (**b**) Pairwise comparisons for ORR; (**c**) Pairwise comparisons for PFS among patients with baseline brain metastasis; (**d**) Pairwise comparisons for PFS among patients without baseline brain metastasis.

**Figure 4 cancers-13-01966-f004:**
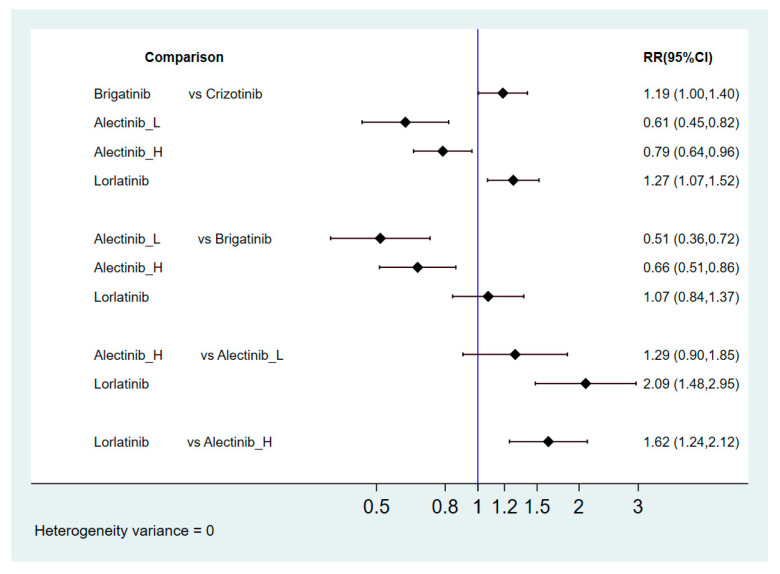
Risk ratio for safety comparisons in grade 3–5 AEs.

**Table 1 cancers-13-01966-t001:** Characteristics of the included studies of first-line ALKI treatment for advanced ALK-positive NSCLC.

Trialr	CROWN		ALTA-1L		ALEX		J-ALEX		ALESIA		Exalt3	
Author	Shaw et al.		Camidge et al.		Camidge et al.		Nakagawa et al.		Zhou et al.		Selvaggi et al.	
Year	2020		2020		2019		2019		2019		2020	
Design	Phase III, open-label, RCT		Phase III, open-label, RCT		Phase III, open-label, RCT		Phase III, open-label, RCT		Phase III, open-label, RCT		Phase III, open-label, RCT	
Intervention	Lorlatinib100 mg QD	Crizotinib250 mg BID	Brigatinib90 mg QD for 7 days, then 180 mg QD	Crizotinib 250 mg BID	Alectinib 600 mg BID	Crizotinib 250 mg BID	Alectinib300 mg BID	Crizotinib250 mg BID	Alectinib600 mg BID	Crizotinib250 mg BID	Ensartinib225 mg QD	Crizotinib250 mg BID
Sample size	149	147	137	138	152	151	103	104	125	62	121	126
Outcome												
PFS(months, median)	NE	9.3	24.0	11.0	34.8	10.9	34.1	10.2	NE	10.7	NE	12.7
PFS (HR, 95% CI)	0.28 (0.19–0.41)		0.49 (0.35–0.69)		0.43 (0.32–0.58)		0.37 (0.26–0.52)		0.37(0.22–0.61)		0.52 (0.36–0.75)	
ORR	113/149	85/147	101/137	85/138	126/152	114/151	76/83	71/90	114/125 *	48/62 *	91/121	85/126
ORR (%)	76%	58%	74%	62%	82.9%	75.5%	92%	79%	91%	77%	75%	67%
Safety												
AE ≥ grade3	72%	56%	73%	61%	45%	51%	26%	52%	29%	48%	NR	NR
Patient characteristics												
Age (median)	59.1	55.6	58.0	60.0	56.3	53.8	61.0	59.5	51.0	49.0	NR	NR
Male (%)	44%	38%	50%	41%	45%	42%	40%	39%	51%	55%	NR	NR
Baseline brain metastasis (%)	26%	27%	29%	30%	42%	38%	14%	28%	35%	37%	NR	NR

Note. NE: not estimable; NR: not reported; * investigator assessed response reported.

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
