# Peer review of "Systematic Review and Network Meta-Analysis of Anaplastic Lymphoma Kinase (ALK) Inhibitors for Treatment-Naïve ALK-Positive Lung Cancer"

_cancers, 2021, doi:10.3390/cancers13081966_

Round 1

Reviewer 1 Report

Thank you for invitation to this reviewing. In this manuscript titled ‘Systemic Review and Network Meta-Analysis of ALK Inhibitors for Treatment-Naïve ALK positive patients’ This study focused on the investigation of therapeutic efficacy of ALK inhibitor and adverse effects using several trials. the design of this study. Many authors explained same contents and there are few readers but know those contents. I think that this manuscript is not suitable for this journal with such a high impact.

Author Response

Response to the Comments from Reviewer #1

  1. Thank you for invitation to this reviewing. In this manuscript titled ‘Systemic Review and Network Meta-Analysis of ALK Inhibitors for Treatment-Naïve ALK positive patients’ This study focused on the investigation of therapeutic efficacy of ALK inhibitor and adverse effects using several trials. the design of this study. Many authors explained same contents and there are few readers but know those contents. I think that this manuscript is not suitable for this journal with such a high impact.

Reply:

Thank you for your comments. First of all, we have to admit that there are some bias and methodological shortcomings in network meta-analysis. However, till now, there are still no head-to-head randomized control trials of new-generation ALK inhibitors due to time and cost concerns in the real world, although comparisons are urgently needed in clinical practice. Meta-analyses and network meta-analyses, as in our work, are widely used to solve problems which may not be directly answered by current randomized control trials. Evidence-based medicine such as network meta-analysis can help clinicians to choose the best treatment and avoid deferring the application of ALK inhibitors in patients with advanced ALK-positive NSCLC. Therefore, in this work, we tried our best to perform a comprehensive network meta-analysis of published RCT trials of ALK-positive NSCLC patients to help clinicians to find the best way to treat such patients.

Reviewer 2 Report

This is a very well written, interesting and timely network meta-analysis on the comparative efficacy and safety of new generation ALK inhibitors in ALK positive NSCLC. Overall the design, methodology , presentation and interpretation of data is sound and the manuscript merits to be published. I have only two comments, the first important:

  1. As this is a network meta-analysis and involves significant statistical considerations and the current reviewer is a medical oncologist specializing in lung cancer, and I am unfamiliar with statistical tools such as SUCRA and ProbBEST, I would suggest review of the manuscript by an expert statistician
  2.  The second paragraph in the discussion session is redundant and repeats knowledge briefly mentioned in the introduction, rendering the whole discussion section too long. I would suggest significantly reducing the size of this paragraph and moving it to the Introduction to merge with the corresponding section

Author Response

Response to the Comments from Reviewer #2

  1. As this is a network meta-analysis and involves significant statistical considerations and the current reviewer is a medical oncologist specializing in lung cancer, and I am unfamiliar with statistical tools such as SUCRA and ProbBEST, I would suggest review of the manuscript by an expert statistician

Reply:

Thank you for your comments. Network meta-analysis is a newly developed methodology which has become increasingly popular in the past decade. Comprehensive statistical analysis was used to achieve indirect comparisons from direct evidence and help clinicians to solve problems that are hard to answer by randomized controlled trials. We believe that the present work is helpful and can benefit ALK-positive NSCLC patients.

  1. The second paragraph in the discussion session is redundant and repeats knowledge briefly mentioned in the introduction, rendering the whole discussion section too long. I would suggest significantly reducing the size of this paragraph and moving it to the Introduction to merge with the corresponding section

Reply:

In order to avoid redundancy and enhance the focus of our study results in the discussion, we have moved the main part of the second paragraph in the discussion to the introduction to provide a more comprehensive background of the evidence and evolution of ALK inhibitors. In addition, this helps the discussion to concentrate on the important findings of our work and on comparisons with other previous meta-analysis. Thanks again for your suggestion. Specifically, we have deleted lines 264-296 in the discussion and merged it into the introduction on lines 93-139.

Reviewer 3 Report

i have attempted to clearly address the most significant issues i found with the manuscript in the attached document. Please accept these as my official critique.

Author Response

Response to the Comments from Reviewer #3

The manuscript is fairly well-written. The plots supplement the text well, with one noted exception (see comment on Page 7). A metaanalysis that uses only 6 studies as source material seems somewhat limited in overall scope. One very interesting result was the finding that one TKI at low-dose was in some ways preferrable to the same drug at high dose. A sort of standard, common phrasing used by many authors is also used here that seems to be out of place in this context. The authors seem to want to justify large trials of TKI’s when what is really needed is a demonstration of how these TKI’s interact with the best available therapies for cancer. That is, unless the authors can recommend clinical trials with immunotherapy that includes a TGF-β blocker and another immunomodulatory drug with a TKI as an adjunct, then they are really suggesting therapy that is sub-optimal for patient healthcare and deleterious for establishing what therapies are best, even if it were the case that only ALK-positive tumors were being treated. TKI’s may also be more appropriate for patients with a history of auto-immune disorders.

Reply:

Immunotherapy is a breakthrough development in oncology and shows promise to greatly improve overall survival in many cancers, including NSCLC. Previous studies have disclosed that NSCLC patients with driver mutations are often less responsive to immunotherapy. However, as per your kind comments, increasing evidence has demonstrated that ALK rearrangements may induce PD-L1 upregulation and create an immunosuppressive tumor microenvironment, which implies that immunotherapy, including Immune checkpoint inhibitors, ALK vaccines, anti-ALK antibodies and Car-T cell therapy, may still play a critical role in the treatment of ALK-positive malignancy, although this is still in phase 1 studies (Emerging Roles of ALK in Immunity and Insights for Immunotherapy, Cancers, 2020). In fact, current phase III RCTs of immunotherapy have always excluded ALK-positive or EGFR-mutated NSCLC because almost no trials have shown a significantly prolonged overall survival with immunotherapy alone or in combination. Furthermore, in some specific patient groups, immunotherapy may be considered as later line therapy, as well established clinical studies including those with low tumor mutation burden have documented in NGS or, as you mentioned, those with autoimmune diseases and intolerance to the side effects of immunotherapy. We conducted this work under the basis of current RCT evidence, and the NCCN guidelines also suggest ALK inhibitors as first-line therapy for patients with advanced ALK-positive NSCLC. Thank you for your kind reminder, we will keep following the progress of immunotherapy in ALK-positive NSCLC and will revise the network meta-analysis based on updated research accordingly. Thank you again for your suggestion.

  1. Lines 44-47:

“Contrast-based analysis was performed for multiple treatment comparisons with the restricted maximum likelihood approach. Treatment rank was estimated using the surface under the cumulative ranking curve (SUCRA), as well as the probability of being the best (Prbest) reference.” Techniques highlighted in yellow may be unfamiliar to many readers and reference to explanatory materials could be helpful. Also, reference to the software used to produce the figures may be of use to some, particularly if it is available as freeware.

Reply:

Detailed information and an additional reference about the statistical methods for network meta-analysis have been provided in the revised version in Materials and Methods: “4.3 Data synthesis and statistical analysis” (Line 394-410). Network meta-analysis (NMA) is a technique used to compare interventions in a single analysis by combining direct and indirect evidence across the network of available studies. One advantage of NMA is that NMA ranks the treatments according to the likelihood (SUCRA was calculated from likelihood). We hope our NMA study provides evidence for clinicians to identify the single best available treatment in terms of efficacy and safety in ALK-positive NSCLC.

  1. Lines 57-58:

“Further head-to-head large-scale phase III RCTs are needed to verify our conclusions.” Personally, I don’t agree. Immunotherapy in combination with a tyrosine kinase inhibitor should be matched against immunotherapy alone.

Reply:

Currently, under the guidance of the NCCN guidelines, advanced ALK-positive NSCLC should generally be treated with a first-line ALK inhibitor. Nevertheless, as per your suggestion, increasing data has focused on the research of the role of immunotherapy in this field. We will update the progress of associated studies on immunotherapy in ALK-positive NSCLC and conduct further research in the future.

  1. Lines 96-100:

“However, patients treated with first-line crizotinib invariably experience progression with secondary ALK gene mutations. The new generation of ALKIs including ceritinib, alectinib, brigatinib and ensartinib have all demonstrated considerable response rates and survival benefits in crizotinib-refractory groups in phase II studies [19- 22].” Both the authors and the oncologists are better off to explain this fact than to simply make note of it. The reason this happens is going to be that some transformed clones are going to adapt to the therapy by acquiring new mutations. These will confer treatment resistance to the old drug, but not to a different treatment regimen.

Reply:

We have revised the section as “Patients treated with first-line crizotinib invariably experience progression due to secondary ALK gene mutations and transformed cancer clones adapting to the therapy, thereby conferring resistance” (Line 107-110) Better illustration on mechanism of secondary mutation was made under such revision. Thank you for kind recommendation.

  1. Page 7 (Sections 2.4.1, 2.4.2 and 2.4.3) I would make just a general remark that these data would perhaps be better presented in tabular format with brief explanatory text.

Reply:

We agree with your concern and have provided the tabular format as league tables in Appendix G. Based on the Preferred Reporting Items for Systematic reviews and Meta-Analyses (PRISMA) statement for network meta-analysis, league tables are used to summarize pairwise comparisons. Additionally, many published studies have used league tables to present the results of pairwise comparisons as well as SUCRA score of each treatment3&4. Please see the table below. Compared to the crizotinib group, the HR of ensartinib was 0.52. The percentage of each treatment (with a green circle) is the surface under the cumulative ranking curve (SUCRA). SUCRA is a numeric presentation of the overall ranking and presents a single number associated with each treatment. SUCRA values range from 0 to 100%, with a higher SUCRA value indicating a higher likelihood that a therapy is in the top rank of the interested outcome. In this table, patients with ALK-positive NSCLC who took lorlatinib had the greatest benefit in PFS and the crizotinib group had the lowest PFS.

  1. Lines 237-239:

“Recently, several next-generation ALKIs such as alectinib, brigatinib, ensartinib and lorlatinib have been developed, and they have different spectrumagainst ALK gene mutations Typographical error.

Reply:

Thank you for pointing this out. We have revised the error. (Line 110)

  1. Lines 276-279:

“In our study, lorlatinib demonstrates the longest PFS with highest SUCRA and Prbest 276 values. However, in pairwise comparisons, lorlatinib was superior in PFS compared to 277 ensartinib and brigatinib, but there was no significant difference betweenlorlatinib, high- 278 and low-dose alectinib.”

Reply:

Thank you for pointing this out. We have revised the error. (Line 314)

  1. Lines 294-297:

“On the other hand, whether the Asian populations in the J-ALEX and ALESIA trials demonstrated better ORR, PFS, OS and CNS response under ALKI is an interesting issue, and the background ALK gene rearrangement variations between Asian and Western populations deserves further investigations.”

Again, I must disagree. The more physiologically important issues to reckon with are immune system function in the treatment of solid tumors.

Reply:

Immune system function indeed plays a critical role in cancer treatment and further investigations of differences in immune systems between Asian and Western populations are important to investigate the different responses to anti-cancer treatment and even side effects. We have revised this section as “On the other hand, whether the Asian populations in the J-ALEX and ALESIA trials demonstrated better ORR, PFS, OS and CNS response under ALKI is an interesting issue, are warranted to investigate the background immune status, tumor microenvironment, and variation of ALK gene rearrangements between Asian and Western populations.” (Line 292-295)

  1. Lines 315-318:

“In addition, low-dose alectinib had a trend of lower risk of AEs than high-dose alectinib though no significant difference in pair-wise comparison, which further emphasizes the urgent need of large-scale RCTs to compare different races and dosages of alectinib.”

I must disagree again on the same grounds a third time. This is suggesting a targeted therapy to treat a large number of patients who would almost certainly be better off if they were all receiving an immunomodulatory regimen (pembrolizumab or ipilumumab, etc.). Enrolling thousands of patients in clinical trials of these drugs has much in common with futile care. These tyrosine kinase inhibitors should be used as second-line and/or adjunct therapy only. What should be suggested, then, is to start with small trials in that context.

Reply:

We agreed with your kind comments. We are eager for trial of combination of Immunomodulaory regimen and TKIs. We will keep updated on the progress of immunotherapy in advanced ALK-positive NSCLC. Furthermore, we will conduct another network meta-analysis of immunotherapy in advanced ALK-positive NSCLC once sufficient trials have been performed.

  1. Lines 407-411:

“Counterintuitive finding is that low dose alectinib demonstrate better survival benefit than high dose alectinib no matter in overall papulation and baseline brain metastasis patient. Of note, all data of low-dose alectinib came from Asian papulation and occult races and genetic factor need to be considered in interpretation of the result.” Grammatical and typographical errors.

Reply:

Thank you for pointing these out. We have revised the errors. (Line 420-422)

Round 2

Reviewer 1 Report

I would like to thank the authors for their efforts improving the manuscript, especially in introduction and discussion. This study focused on the therapeutic effectiveness of ALK inhibitors and adverse effects using several clinical trials. I think that many authors published same contents and this manuscript only provided what I have the knowledge already known. I do not think that this manuscript is suitable for this journal.